# Multiparametric Investigation of Welding Techniques on Toe Radius of High Strength Steel at Low-Temperature Levels Using 3D-Scanning Techniques

**Miroslav Randić** [1,*], **Duško Pavletić** [2] and **Goran Turkalj** [3]

[1] Branch Office Rijeka, Croatian Register of Shipping, Rijeka 51000, Croatia
[2] Department of Industrial Engineering and Management, Faculty of Engineering, University of Rijeka, Rijeka 51000, Croatia; duskop@riteh.hr
[3] Department of Engineering Mechanics, Faculty of Engineering, University of Rijeka, Rijeka 51000, Croatia; goran.turkalj@riteh.hr
* Correspondence: miroslav.randic@crs.hr; Tel.: +385-98-328-603

**Abstract:** To avoid the occurrence of surface cracks at the welds, it is important to lower the stress concentration in the zone of the weld face by an appropriate choice of parameters. A plethora of experiments was conducted varying four welding techniques. The welded samples were scanned with 3D scanners and the toe radius was measured on each sample. The significance of the obtained results was analyzed using Pareto diagrams. The experiment results analysis shows that the length of the electrode stick-out has a significant influence on the toe radius, while the shielding gas has a great effect on the toe radius. Moreover, with the analysis of results obtained by experiments it was proved that the interaction of the torch angle and the number of cover passes, as well as that of the torch angle and the shielding gas, has a significant influence on the toe radius.

**Keywords:** welded joint; geometric stress concentration factor; toe radius; welding techniques; 3D-scanner; sensitivity analysis; butt-welded joint

---

## 1. Introduction

Welding is still the most common method of joining metal in industrial production. In exploitation, welded constructions, as well as joints, are often affected by cyclic loading. Such loadings facilitate a crack initiation [1] and its propagation [2], often in the welded section [3]. The initiation of cracks in a welded joint occurs in places of the greatest stress concentration, i.e., in the toe radius. The aim of this study was to determine the influence of different welding techniques on the toe radius. Furthermore, the objective of the choice of favorable welding techniques that were preformed was to enable the formation of a welded joint surface onto which surface cracks will not be initiated. Figure 1 shows a welded joint, in exploitation for approximately five years, with a visible crack initiated at the weld toe. The weld is on the main deck of a ship that transports oil products, that is, the place where the highest tensile stresses occur [4]. Thus, it is of great importance to obtain quality welds in which cracks will not initiate [5].

The irregular weld surface contributes to the cracks initiation [6]; namely, it is the geometrical characteristics of the surface that effect the local stress concentration factor [7]. Cracks most frequently initiate in places of the highest tensile stress, which is the area of the toe [8]. The stress concentration factor increases by decreasing the toe radius [9], which can be calculated by empirical expressions [10]. In practice, there are several techniques to form the weld surface in order to lower the stress concentration [11]. The grinding technique is most frequently used on ships that are being built [12].

Cozzolino et al. [13] recommended post-weld rolling methods to reduce stresses, while Haagensen and Maddox [14] recommended several methods on the post-weld improvement of the weld, such as burr grinding, tungsten inert gas (TIG) dressing, hammer peening, and needle peening; however, all the approaches that are carried out after the completion of welding require additional resources and increase in the production costs. The stress concentration factor in the butt-welded joint depends on the weld shape [15]. The primary hypothesis was to achieve the geometry of the welded joints with a lower sensitivity to the initiation of surface cracks by selecting the right choice of parameters.

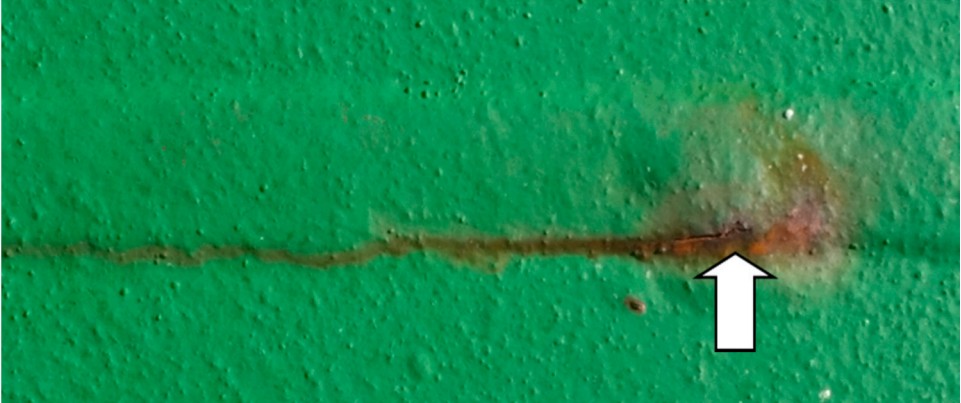

**Figure 1.** Crack at the weld toe.

The geometry of the welded joint can be changed by exchanging welding techniques [16], as well as by changing welding parameters [17]. Bytyqi et al. [18] analyzed the effects of welding current on the weld shape and established that the weld width increases by the increase of the welding current. Kasuya and Yurioka [19] researched the effects of the surrounding temperature on the crack initiation in the welded joint. Nouri et al. [20] examined the effects of the wire speed, welding speed, and the wire stick out on the weld geometry. Moreover, Tewari et al. [21] researched the effects of welding speed and the heat input on the penetration depth.

The effect of the weld geometry on the weld stress concentration was analyzed by Williams et al. [22], while the effect of the weld toe angle was analyzed by Robakowski [23]. In these analyses, it was established that the toe radius and the weld toe angle have influence the stress concentration. Berge [24] used a geometric model to describe the effect of the change of the base material thickness on the stress concentration in the toe radius of the butt-welded joint. Provided that the nominal stress in the thicker and thinner base material is the same, the stress gradient reduces from the maximum to the nominal value in the thinner plate more than in the thick plate. Using the finite element method (FEM), Cerit et al. [25] carried out the research on the effect of the weld toe angle on stress concentration, with regard to the toe radius and thickness of the base material. In their studies, they concluded that the stress concentration factor acquires values higher than three when the weld toe angle increases to 60°, and the toe radius decrease to 0.5 mm. Kim et al. [26] analyzed the way in which additional material widens over the base material, which depends on the welding current voltage, the diameter of additional material, and on the speed of weld passes, concluding that the width of the welded joint increases significantly with the increase of the diameter of the additional material wire.

Figure 2 shows five geometric quantities that have an impact on the stress concentration factor in the welded joint, i.e., $\phi$—toe radius; $\theta$—weld toe angle; $W$—weld width; $t$—thickness of the base material, and $h$—reinforcement height. In the literature, there are a few empirical expressions for the calculation of the stress concentration factor in a butt-welded joint, which use the aforementioned geometric quantities [10,27]. As all geometric quantities do not have equal influence on the stress concentration, the sensitivity analysis of the expression proposed by Ushirokawa and Nakayama [28] were carried out in order to determine which geometric quantity has the biggest influence on the stress concentration factor [29,30]. This analysis can be found in the Appendix A.

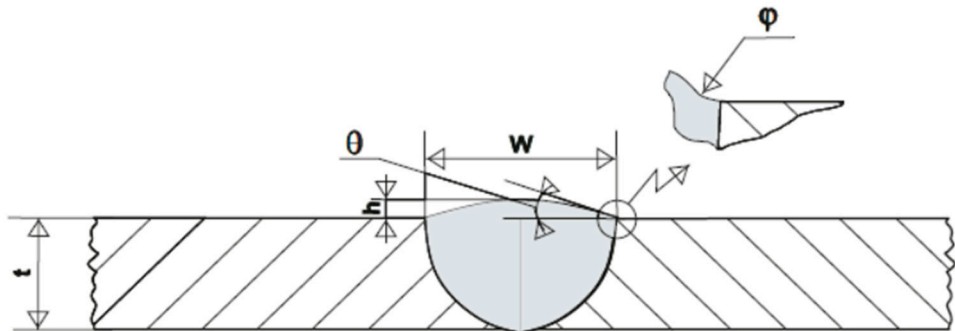

**Figure 2.** Influential geometric quantities of the butt-welded joint.

In the present study, the toe radius was scanned by the 3D technique with utmost precision. After that, the toe radius was measured using state-of-the art computer techniques. These data were then used for further processing. By the comparative study of the toe radii measurements, the influence of four welding techniques was obtained. The interaction of welding techniques on the toe radius was obtained by multiparametric investigation of welding techniques using the Pareto and the influence diagram. The following welding techniques were chosen for this analysis: Torch angle, a number of cover passes, length of electrode stick-out, and shielding gas. It is these welding techniques that were chosen as, the research to date has not sufficiently investigated these details enough. The state of the art 3D scanning of the welded joint and the computational measuring of the welded joint geometry have contributed to the precision of this research.

This paper describes 24 experiments performed using four different welding techniques (i.e., torch angle, number of cover passes, length of electrode stick-out, and shielding gas), which might significantly affect the weld face geometry of high strength steel at low temperatures (EH36), and prevent the weld surface cracks during shipbuilding. Previous studies have reported that the seam edge radius is a geometric size that affects the stress concentration and the initiation of surface cracks. Therefore, the focus in this paper was to examine the seam edge radius; the way the seam edge radius is measured, and the ways in which the maximum seam edge radius can be achieved during welding. Furthermore, the surface of these welds was scanned with the latest 3D scanning technique and further suggested several new welding parameters to avoid the initiation of surface cracks.

In conclusion, new welding parameters have been recommended for achieving favorable shape of the welded joint surface, thus preventing possible surface cracks.

## 2. Materials and Methods

The V-butt joint in high strength steel at low temperatures has been used in the research. According to the shipping classification society Det Norske Veritas/Germanischer Lloyd (DNV GL) [31], the steel was marked as VL E36. The thickness of the base metal is 10 mm. The welding was performed according to the approved welding procedure specification (WPS) for the two shielding gases that were used [32].

A total of twenty-four experiments were performed, in which the four welding techniques were systematically changed by the principles of full factorial design of experiments.

Table 1 shows the welding techniques that were changed during the experiment. The torch angle was analyzed in three levels (forehand technique, vertical technique, and backhand technique). The number of cover passes was analyzed in two levels (one pass and three passes). Furthermore, the length of the electrode stick-out was analyzed in two levels (5 mm and 15 mm); the experiments were performed with two types of shielding gas "−" mixture (82% Ar and 18% $CO_2$) and "+" 100% $CO_2$.

**Table 1.** Welding techniques that were changed during the experiment.

| Welding Techniques | Level | | |
|---|---|---|---|
| | Lower (Mark "−") | Middle (Mark "0") | Higher (Mark "+") |
| Torch angle (A) | Forehand technique Group 1 | Vertical technique Group 2 | Backhand technique Group 3 |
| Number of cover passes (B) | 1 pass | | 3 passes |
| Length of electrode stick-out (C) | 5 mm | | 15 mm |
| Shielding gas (D) | 82% Ar + 18% $CO_2$ | | 100% $CO_2$ |

In Table 2, the full factorial design of the experiments is given. The length of the sample used is 150 mm. The experiments were divided into three groups. The groups are labeled with numbers one, two, and three, each one presenting a different torch angle, as shown in Table 1. In each group, eight experiments were made, labeled by numbers from one to eight. These stand for the other three welding techniques that were varied by the method of the full factorial design of experiments.

**Table 2.** Design of the experiments.

| Input Factor | The Experiment Group and Label Level of Input Factor | | | | | | | | | | | | | | | | | | | | | | | |
|---|---|---|---|---|---|---|---|---|---|---|---|---|---|---|---|---|---|---|---|---|---|---|---|---|
| | 1-1 | 1-2 | 1-3 | 1-4 | 1-5 | 1-6 | 1-7 | 1-8 | 2-1 | 2-2 | 2-3 | 2-4 | 2-5 | 2-6 | 2-7 | 2-8 | 3-1 | 3-2 | 3-3 | 3-4 | 3-5 | 3-6 | 3-7 | 3-8 |
| | Group 1 | | | | | | | | Group 2 | | | | | | | | Group 3 | | | | | | | |
| Torch angle | − | − | − | − | − | − | − | − | 0 | 0 | 0 | 0 | 0 | 0 | 0 | 0 | + | + | + | + | + | + | + | + |
| Number of cover passes | − | − | − | − | + | + | + | + | − | − | − | − | + | + | + | + | − | − | − | − | + | + | + | + |
| Length of electrode stick-out | − | − | + | + | − | − | + | + | − | − | + | + | − | − | + | + | − | − | + | + | − | − | + | + |
| Shielding gas | − | + | − | + | − | + | − | + | − | + | − | + | − | + | − | + | - | + | − | + | − | + | − | + |

## 2.1. Materials

In this study, the base material grade VL E36 was used. This material is increasingly used in the shipbuilding industry when new ships are constructed. In Table 3 shows the required chemical composition of the VL E36 according to the rules of the DNV GL [31]. It should be emphasized here that such a chemical composition corresponds to the certificate issued by Det Norske Veritas.

**Table 3.** Chemical composition of the base material used in the research.

| Chemical Element (%) | C | Si | Mn | P | S | Cr | Mo | Ni | Al | Cu | Nb |
|---|---|---|---|---|---|---|---|---|---|---|---|
| Required (according to DNV GL Rules) [31] | ≤0.18 | ≤0.50 | 0.90–1.60 | ≤0.035 | ≤0035 | ≤0.20 | ≤0.08 | ≤0.40 | ≥0.020 | ≤0.35 | 0.02–0.05 |
| Actual (according to factory certificate) [33] | 0.176 | 0.34 | 1.42 | 0.014 | 0.001 | 0.050 | 0.003 | 0.020 | 0.024 | 0.020 | 0.026 |

## 2.2. Varying Welding Techniques

Four welding techniques were used during welding:

- torch angle;
- number of cover passes;
- the length of electrode stick-out;
- the shielding gas.

### 2.2.1. Torch Angle

A torch angle is an angle between the torch and the weld molten pool. This welding technique was varied in three levels during the experimental investigation. The effect of the latter welding technique on the toe radius has not been investigated according to the available literature.

### 2.2.2. Number of Cover Passes

Welding can be performed with a small or a large number of cover passes. This welding technique effects the welded joint surface. Yet, so far, no studies have been reported on this welding technique; only few case studies from practice are available.

### 2.2.3. Length of Electrode Stick-Out

Length of electrode stick-out is the length of a wire coming out from the contact tip of the torch measured along the welding wire. According to the available literature, the influence of this welding technique on the toe radius has not been reported so far.

### 2.2.4. Shielding Gas

Shielding gases that are used in electric arc welding can be divided into two main groups, that is, active gases and inert gases. The most commonly used active gas is carbon dioxide $CO_2$, while argon is the most commonly used inert gas. Over the recent times, mixtures of gases that have better characteristics than pure gases are used ever more frequently. The welding procedure is named after the type of gas used. Two shielding gases were used in this research, i.e., a mixture of gases (82% Ar and 18% $CO_2$) and pure $CO_2$.

## 3. Results

### 3.1. D-Scanning and Methodology

Upon welding, the surface of each sample was scanned with a three-dimensional scanner in the Centre for Advanced Computing and Modelling of the University in Rijeka (Figure 3) [34].

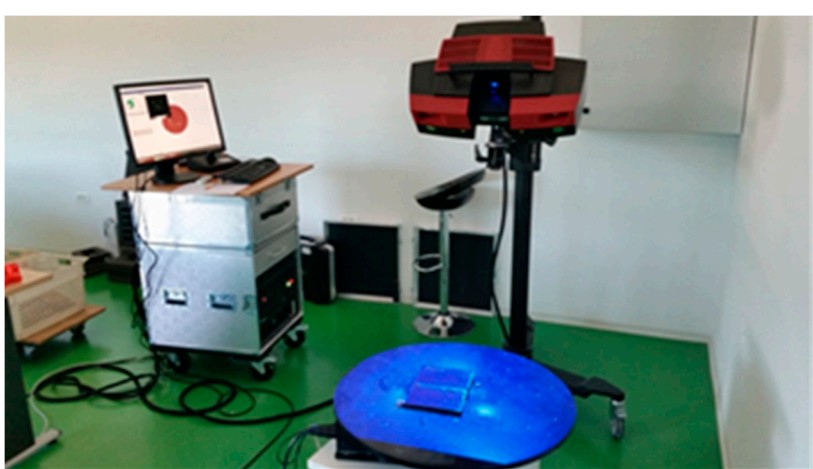

**Figure 3.** Scanning of the samples by 3D scanner.

The ATOS II Triple Scan system (GOM GmbH, Braunschweig, Germany) was used for scanning, consisting of two cameras, a projector, and a control unit.

The projector is used for emitting narrowband blue light that enables precise measurements and digitalization independently of environmental lighting conditions. The projected blue light is reflected from the object, and is measured using two cameras with a resolution of $5 \times 10^6$ pixels each. According to the instructions of GOM Inspect computational software (GOM GmbH, Braunschweig, Germany), the minimum distance between two points on the sample measured was 0.02 mm [35].

The digitalized surface of each sample was imported into the GOM Inspect computational software, as a result of which weld geometries of the welded joint were measured. GOM Inspect computational software is a computational software developed for the analysis of data obtained by three-dimensional scanning.

The toe radius of each sample was measured using three bands. The first band was determined in the area of 20 to 30 mm after the commencement of welding; the second band was placed in the center of the sample, 70 to 80 mm from the commencement of welding; and the third band was placed in the area of 120 to 130 mm from the commencement of welding. During welding, the automatic reading of welding parameters (the strength of welding current, the power of welding current, and the wire

feed speed) was preformed. It was established that welding parameters became stabilized prior to the initial band in which welded joint geometries were measured.

Each band was 10 mm wide, while the distance between cross sections was 1 mm. Figure 4a shows the sample generated in the GOM Inspect with bands in which cross-section geometries were measured, while in Figure 4b the actual samples with marked bands are presented. Furthermore, Figure 5 shows the transverse appearance of a cross-section; in this case, a cross-section of 20 mm is distanced from the beginning point of the welding.

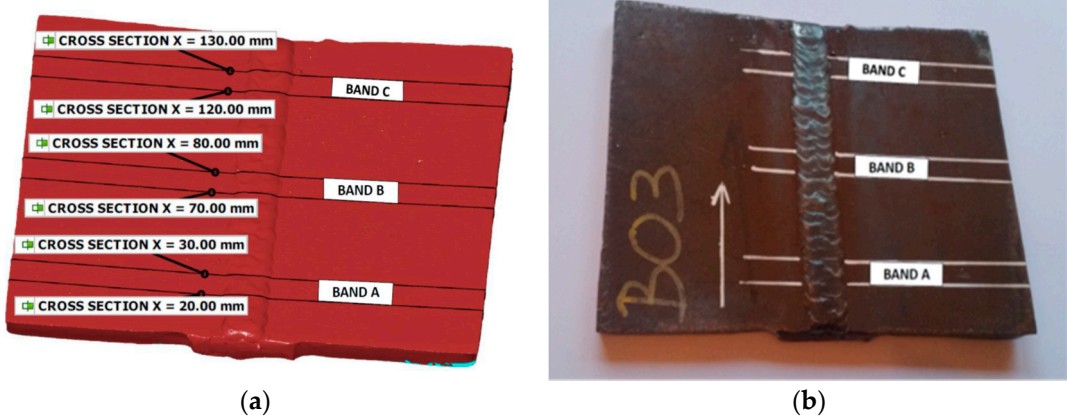

(**a**)  (**b**)

**Figure 4.** The sample: (**a**) Generated in the GOM Inspect computational software; (**b**) with marked bands.

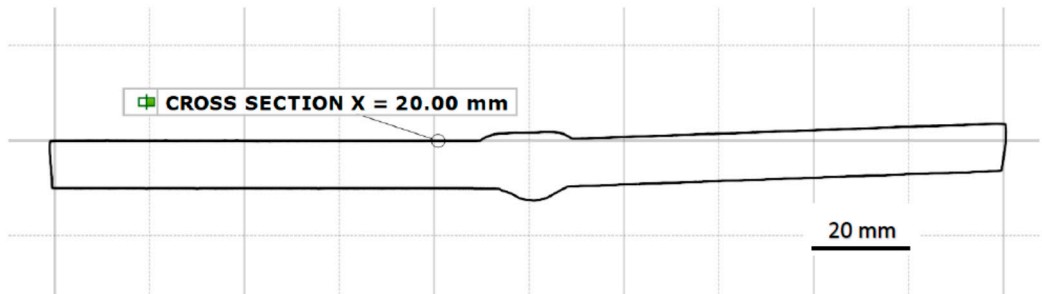

**Figure 5.** Graphical representation of cross section X = 20.00 mm.

Each cross-section was geometrically processed separately; the measurement of the toe radius was performed from the left and the right side of the welded joint.

*3.2. Measurement of Geometries of the Welded Joint Cross-Section With the GOM Inspect Computational Software*

As it was mentioned above, three transverse bands were singled out on each sample, and on each band measurements, eleven cross-sections were made.

For each sample, toe radius was measured at 33 cross-sections, using two positions at each cross-section (left and right positions). A total of 66 measurements at each sample were analyzed. One example for sample two to three, band A, cross-section 20 mm was described. At each cross-section a point was established in which the base material surface extends into the weld reinforcement height. For the left side of the weld, this point is marked POINT LEFT 02, Figure 6. On all cross-sections, this point is marked at the point where the weld reinforcement height begins.

Toe radius can be defined as the radius of a circle that passes through three points on the weld surface. In that case, the value of the toe radius depends on the distance between the points through which the circle passes. The distance of 0.125 mm between points was chosen, following the research of Lawrence et al. [8], in which it was established that the most significant stress concentration occurs in the very area of 0.125 mm along the welded joint.

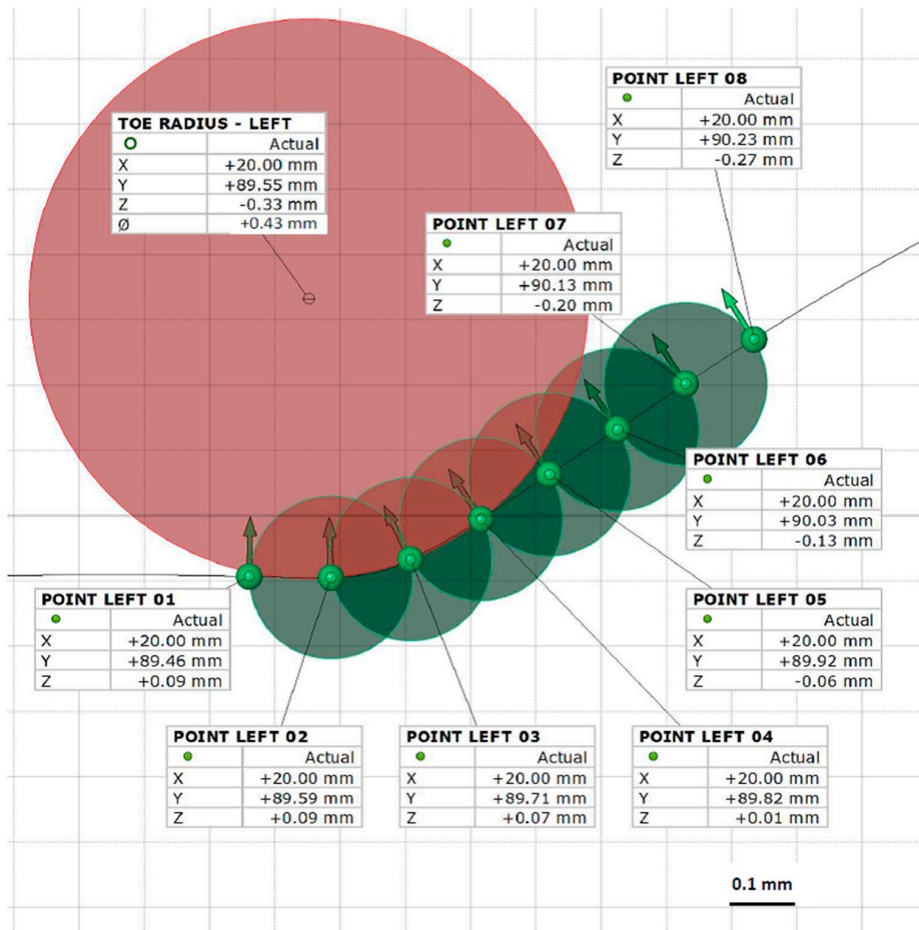

**Figure 6.** Weld toe radius in the GOM Inspect.

When measuring the weld toe radius using the GOM Inspect software package, circles were constructed through each of the three marked points on the weld surface. As the stress concentration is higher in the area of the smallest toe radius, the circle with the smallest radius is evaluated in the further analysis. In the case shown in Figure 6, these are the POINT LEFT 02, POINT LEFT 03, and POINT LEFT 04, while the circle that passes through these three points is marked with red color. In this example, the circle radius is 0.43 mm.

Table 4 presents the smallest measured toe radii in each band.

**Table 4.** Values of the smallest measured toe radii for each sample in millimeters (mm).

| Sample | 1-1 | 1-2 | 1-3 | 1-4 | 1-5 | 1-6 | 1-7 | 1-8 | 2-1 | 2-2 | 2-3 | 2-4 |
|--------|-----|-----|-----|-----|-----|-----|-----|-----|-----|-----|-----|-----|
| Band A | 0.22 | 0.27 | 0.27 | 0.30 | 0.20 | 0.43 | 0.22 | 0.36 | 0.29 | 0.27 | 0.33 | 0.29 |
| Band B | 0.21 | 0.41 | 0.29 | 0.30 | 0.29 | 0.39 | 0.26 | 0.31 | 0.27 | 0.40 | 0.26 | 0.26 |
| Band C | 0.23 | 0.30 | 0.26 | 0.23 | 0.29 | 0.29 | 0.26 | 0.45 | 0.41 | 0.40 | 0.28 | 0.24 |
| **Sample** | **2-5** | **2-6** | **2-7** | **2-8** | **3-1** | **3-2** | **3-3** | **3-4** | **3-5** | **3-6** | **3-7** | **3-8** |
| Band A | 0.39 | 0.28 | 0.27 | 0.26 | 0.48 | 0.38 | 0.37 | 0.30 | 0.28 | 0.31 | 0.26 | 0.35 |
| Band B | 0.39 | 0.24 | 0.28 | 0.33 | 0.29 | 0.33 | 0.25 | 0.29 | 0.27 | 0.29 | 0.21 | 0.25 |
| Band C | 0.37 | 0.25 | 0.25 | 0.29 | 0.37 | 0.33 | 0.32 | 0.28 | 0.25 | 0.35 | 0.31 | 0.36 |

The obtained results were analyzed by the method of design of experiments [36]; the effect of each of the monitored welding techniques on the toe radius of the weld was estimated [37]. With the optimal choice of the values of welding techniques, it is possible to obtain the toe radius with the minimal initiation of surface cracks.

The analysis of the results obtained by the experiments and their interpretation is essential for drawing conclusions that can be of use in engineering practice. The first step is to identify welding techniques and their interaction, that is, those that have a significant effect on the toe radius (Table 5). In the farthest right column of this table, the *p*-value is shown, the value based on which it is established how significant specific data is for the experiment results. The 95% significance level is commonly taken [38].

**Table 5.** Table of influence of main effects and their interactions.

| | Main Effects and Interactions | *p*-Value |
|---|---|---|
| A | Torch angle | 0.267 |
| B | Number of cover passes | 0.783 |
| C | Electrode stick-out | **0.036** |
| D | Shielding gas | 0.053 |
| AB | Torch angle—Number of cover passes | **0.016** |
| AC | Torch angle—Length of electrode stick-out | 0.366 |
| AD | Torch angle—Shielding gas | **0.027** |
| BC | Number of cover passes—Length of electrode stick-out | 0.342 |
| BD | Number of cover passes—Shielding gas | 0.247 |
| CD | Length of electrode stick-out—Shielding gas | 0.882 |

The conclusion drawn from the table is that effect C (length of electrode stick-out), as well as interactions AB (torch angle—number of cover passes) and AD (torch angle—shielding gas) have a significant influence on the toe radius. These values are shown in bold. This is also shown in the Pareto diagram given in Figure 7.

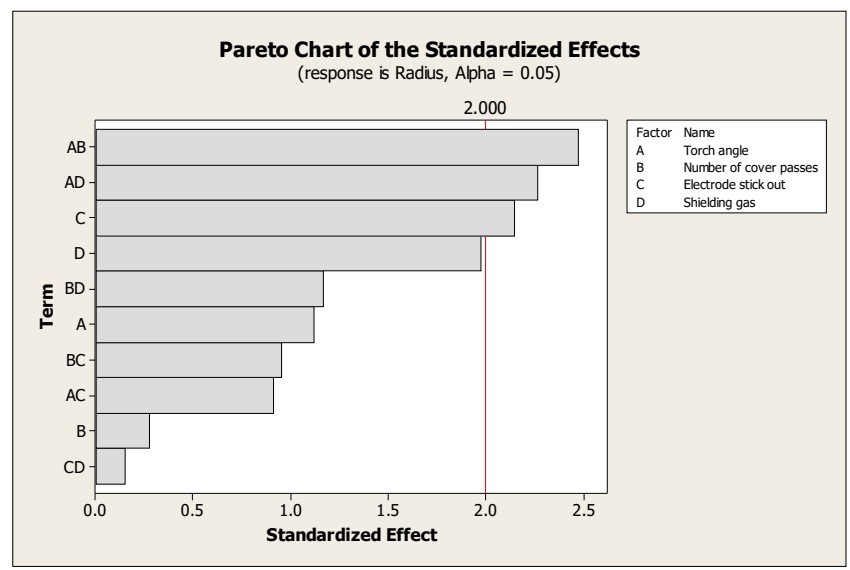

**Figure 7.** Influence of different parameters using Pareto diagram.

The effect of D (shielding gas) is great; however, with regard to the significance level of 95%, this effect is not significant.

The influence of the main effects is shown in Figure 8a and their interactions is shown in Figure 8b. The charts presented to confirm and complete the conclusions from Table 5. The line that shows the effect of the electrode stick-out is the most vertical, its influence on the toe radius being the biggest and significant. Small toe radii occur in bigger lengths of the electrode stick-out. The line that shows the influence of the shielding gas is vertical, however, according to Table 5, this influence can be considered as big, though not significant. The influence of the number of cover passes is negligible, while the influence of the torch angle is small. The charts in Figure 8b show double-sided interactions

of individual main effects. The lines that are parallel between themselves, present a smaller interaction. It is possible to notice that the lines of the interaction of main effects, namely, the electrode stick-out (C) and the shielding gas (D), are almost parallel, which means that their interaction is negligible. The charts that present the interaction of the torch angle (A) and the number of cover passes (B), as well as the torch angle (A) and the shielding gas (D), are intersected in a big angle, which means that their interaction is significant. The other three interactions are big, however, they are not significant.

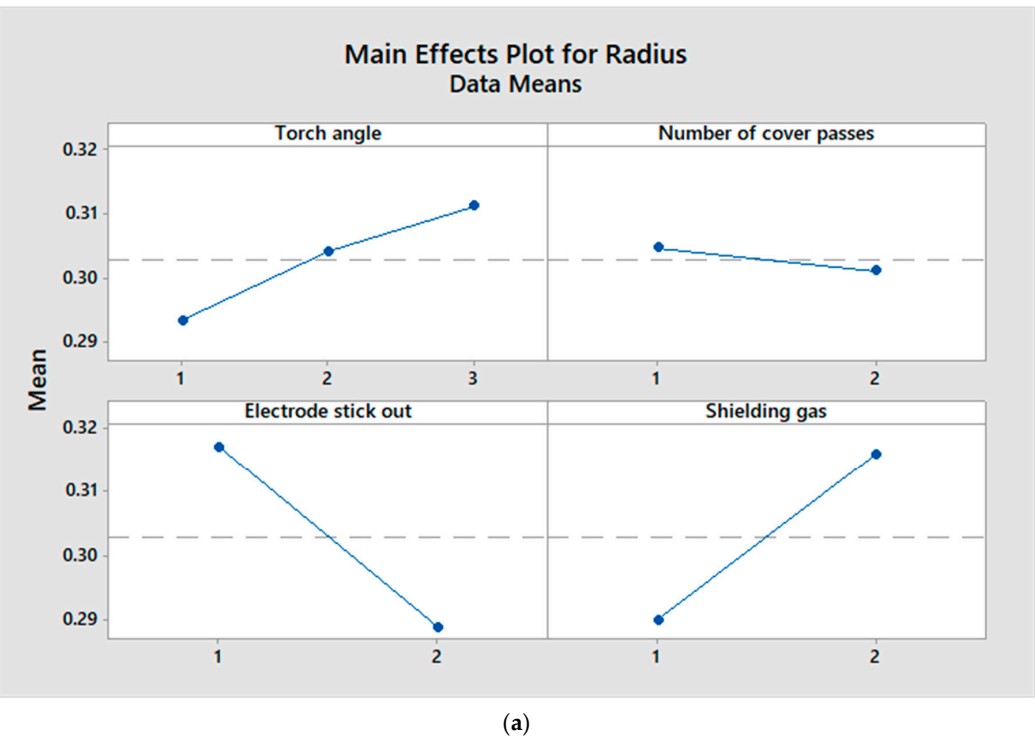

(a)

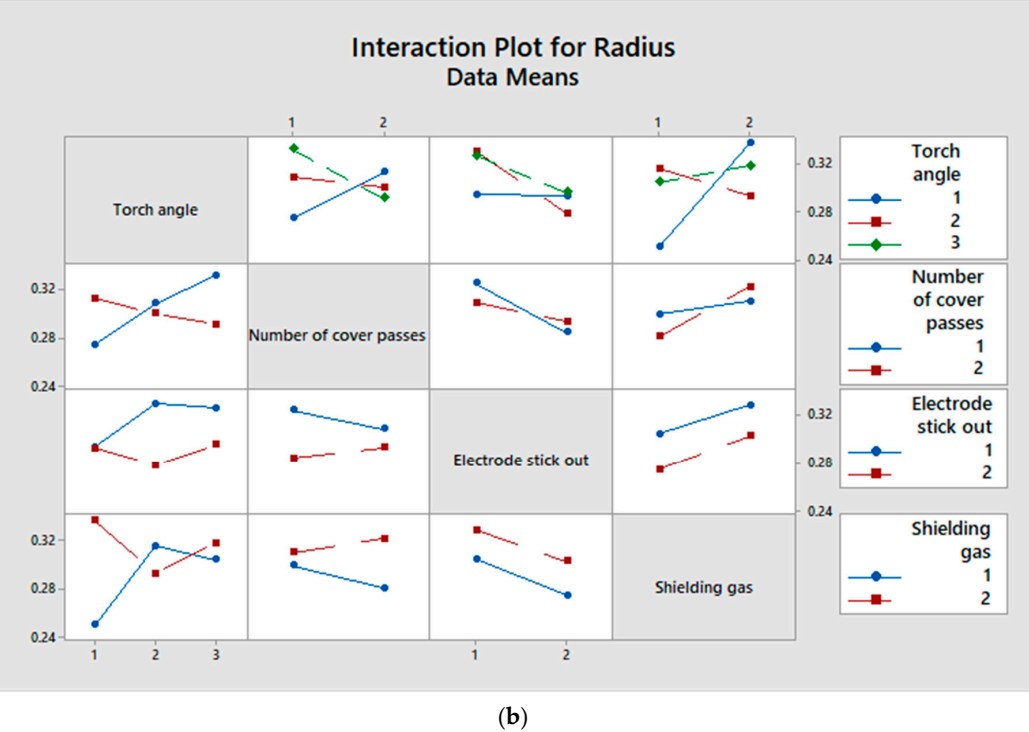

(b)

**Figure 8.** (**a**) Influence of main effects, (**b**) interactions of main effects.

Contour graphs presented in Figure 9 shows the relationship of the electrode stick-out and the torch angle for different values of the number of cover passes and the shielding gas. By looking at the chart, it can be seen that the higher gradient of the radius change is for one cover pass and the shielding gas mixture (Figure 9a). In that case, the smallest radii occur with the length of the electrode stick-out of 15 mm and the forward welding technique, while the number of cover passes is one and the mixture (82% Ar and 18% $CO_2$) of shielding gas is used, marked with "A" in Figure 9a. This welding technique should be avoided.

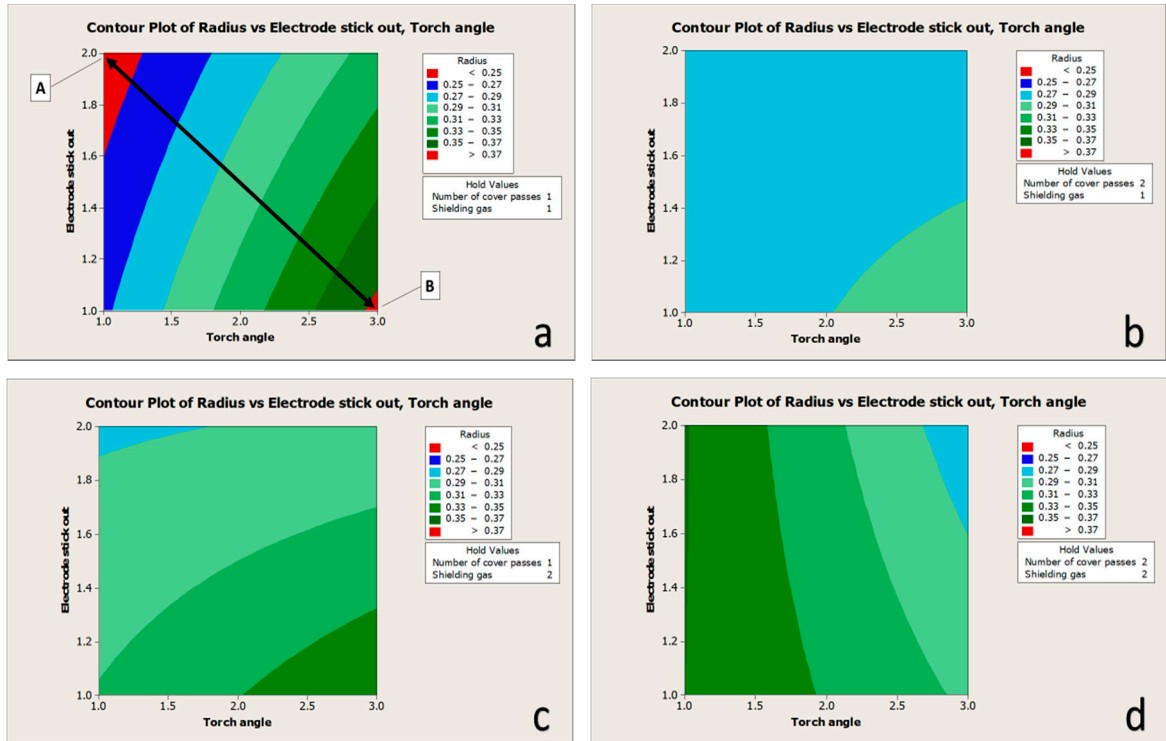

**Figure 9.** Calculation of the toe radius as a function of the torch angle and the electrode stick-out for following welding techniques (**a**) one cover pass; shielding gas 82% Ar + 18% $CO_2$, (**b**) three cover passes; shielding gas 82% Ar + 18% $CO_2$, (**c**) one cover pass; shielding gas 100% $CO_2$ and (**d**) three cover passes; shielding gas 100 $CO_2$.

In the opposite angle of the same chart, the higher radii occur (>0.37 mm), marked with "B" on the chart. Thus, according to the implemented experiments, it is when welding with the small electrode stick-out, backhand welding technique, with one cover pass and using a mixture of shielding gases that we can expect the biggest radius in which the fewest surface cracks are initiated.

## 4. Discussion

These investigations were performed on butt-welded joints of high strength steel at low temperature; yet, this procedure can be also applied to other types of steel, while its application it is not recommended for aluminum. The results of these investigations can be used on radii from 0.20 mm to 4 mm, i.e., those obtained during investigations. Their application on other radii is not recommended.

These results are applicable for a maximum of three cover passes. With a higher number of cover passes the weld surface takes a different form so that the results obtained in that way can depart from those obtained in this research.

In the present study, four welding parameters were changed, and different weld radius of the weld joint was obtained. The key findings of this paper are the following:

(1)　　a new state-of-the-art method for scanning a welded joint surface and measuring the seam edge radius with a 3D scanner have been presented;

(2)　　by analyzing the radius of the seam edge with a 3D scanner, new welding parameters were obtained which significantly affect the surface shape of the welded joint, thus preventing the initiation of surface cracks;

(3)　　welding parameters have been obtained which should definitely be avoided during welding because if used, there is a high probability of surface cracks.

In this paper, we have discovered a new way to scan the weld surface, thus obtaining a more accurate welded joint radius. This method can be used for any welded joint, independently of the material. The only disadvantage of this method is that it is very slow and unsuitable for scanning welds except under laboratory conditions.

## 5. Conclusions

The influence of four techniques of welding on the welded joint surface, i.e., on the toe radius, was analyzed in the research. The following techniques were analyzed: Torch angle, the number of cover passes, the length of the electrode stick-out, and shielding gas. Twenty-four experiments were conducted varying these four welding techniques. The welded samples were scanned with 3D scanners and the toe radius was measured on each sample.

The experiment results analysis shows that the length of the electrode stick-out has a significant influence on the toe radius. Smaller toe radii occur when the electrode stick-out is longer.

There is a high, influence of the shielding gas, namely, it was established that with the mixture of shielding gases smaller toe radii occur. The torch angle also has a big influence, while the number of cover passes has a negligible influence on the toe radius.

With the analysis, it was established that interactions of the torch angle and the number of cover passes (AB), as well as that of the torch angle and the shielding gas (AD), are significant. The toe radius significantly changes if the torch angle and number of cover passes change, as well as the shielding gas, especially so when the forward welding technique is used.

The smallest toe radii occur with the longer electrode stick-out, the forehand welding technique with one cover pass and welding with a mixture of shielding gases. Thus, if the propagating of cracks on the toe radius is to be avoided, the aforementioned welding techniques definitely need to be avoided. The optimal welding technique depends on the selection of the shielding gas used. With a mixture of shielding gases (82% Ar and 18% $CO_2$) the torch angle should be perpendicular, while with the pure $CO_2$ the torch has to be in the forehand position. In both cases, the electrode stick-out should be short, while the number of cover passes is not a parameter that significantly influences the stress concentration and can be independently selected.

To avoid the occurrence of surface cracks at the welds, it is important to lower the stress concentration in the zone of the weld face by an appropriate choice of parameters. The results of this research will enable the welding engineering a more appropriate selection of welding technology for the welding of high-strength steel used at a low temperature. The selected technology should prevent or minimize crack initialization at welded joints during the exploitation.

**Author Contributions:** Conceptualization, M.R., D.P. and G.T.; methodology, M.R., D.P. and G.T.; software, M.R.; validation, M.R., D.P. and G.T.; formal analysis, M.R., D.P. and G.T.; investigation, M.R.; resources, M.R.; data curation, M.R.; writing—original draft preparation, M.R.; writing—review and editing, D.P. and G.T.; visualization, M.R.; supervision, D.P.; project administration, M.R.; funding acquisition, D.P.

**Funding:** This work has been supported by the Croatian Science Foundation under the project IP-2018-01-3739 and the University of Rijeka (contract nos. uniri-tehnic-18-33 and uniri-tehnic-18-107).

**Acknowledgments:** The authors acknowledge the Centre for Advanced Computing and Modelling (CNRM) for providing scanner "ATOS II Triple Scan" at the University of Rijeka in Rijeka, Croatia. The scanner and other ICT research infrastructure were acquired through the project "Development of Research Infrastructure for Laboratories of the University of Rijeka Campus", which is co-funded by the European Regional Development Fund. The support of Vedran Margan from CNRM to the technical work is greatly acknowledged.

**Conflicts of Interest:** The authors declare no conflict of interest.

**Appendix A**

*The Sensitivity Analysis of Expression for Calculation of Stress Concentration Factor in Butt Welding Joints*

The stress concentration factor is the ratio of maximum stress, $\sigma_{max}$ and nominal stress, $\sigma_{nom}$, i.e.,

$$K = \frac{\sigma_{max}}{\sigma_{nom}} \tag{A1}$$

The surface of the weld is irregular with a number of local form changes that have a significant impact on the value of the geometric stress concentration factor [39]. Ushirokawa and Nakayama [28] suggested the expression for the calculation of the geometric stress concentration factor at the butt-welded joint in the following form:

$$K_t = 1 + \left( \frac{1 - e^{-0.90 \cdot \theta \cdot \sqrt{\frac{t+2h+0.6W}{2h}}}}{1 - e^{-0.90 \cdot \frac{\pi}{2} \cdot \sqrt{\frac{t+2h+0.6W}{2h}}}} \right) \cdot 2 \cdot \left[ \left( \frac{h}{\varphi} \right) \cdot \frac{1}{2.8 \cdot \frac{t+2h+0.6W}{t} - 2} \right]^{0.65} \tag{A2}$$

where: $\varphi$ is the toe radius; $\theta$ is the weld toe angle; $W$ is the weld width; $t$ is thickness of the base material; while $h$ denotes the reinforcement height.

From Equation (A2) it can be concluded that each of the five geometric quantities does not influence the stress concentration factor in the same way. Therefore, the sensitivity analysis is carried out in order to determine the level of impact of each single geometric quantity on the stress concentration factor [29].

Two models of the sensitivity analysis were carried out. A partial derivation of the expression (A2) with respect to each geometric quantity was carried out in model A. In model B, two geometric quantities that have the highest impact on the stress concentration factor were first singled out, then the expression (A2) was analyzed with respect to these two geometric quantities.

Model A—Complete Derivation of the Expression

The sensitivity analysis is carried out by the partial derivative of Expression (2) with respect to each geometric quantity, i.e.,

$$U_\theta = \left( \frac{\partial K_t(t, \theta, \varphi, h, W)}{\partial \theta} \right) \tag{A3a}$$

$$U_t = \left( \frac{\partial K_t(t, \theta, \varphi, h, W)}{\partial t} \right) \tag{A3b}$$

$$U_\varphi = \left( \frac{\partial K_t(t, \theta, \varphi, h, W)}{\partial \varphi} \right) \tag{A3c}$$

$$U_h = \left( \frac{\partial K_t(t, \theta, \varphi, h, W)}{\partial h} \right) \tag{A3d}$$

$$U_W = \left( \frac{\partial K_t(t, \theta, \varphi, h, W)}{\partial W} \right) \tag{A3e}$$

In Equations (A3a)–(A3e), the partial derivative describes the rate of change of the function regarding the change of the independent variable. The positive value of the derivative expresses the positive sensitivity of the independent variable to the result [30].

For each influential geometric quantity, the span of expected quantity value has been determined first and the sensitivity analysis for values within that area is carried out. For the two geometric quantities with the highest impact (the weld toe angle and the toe radius), the sensitivity analysis is carried out for four values, while for the other three geometric quantities it is carried out for three values, Table A1.

**Table A1.** The area of sensitivity analysis and of values in which the sensitivity is carried out.

| Geometric Quantity | Values in which the Sensitivity was Analyzed |
|---|---|
| Thickness of the base metal | 6, 10, and 20 mm |
| Weld toe angle | 5°, 10°, 20°, and 50° |
| Toe radius | 0,1, 0.5, 1, and 3 mm |
| Reinforcement height | 1, 2, and 4 mm |
| Weld width | 20, 25, and 30 mm |

Once the sensitivity analysis has been performed, four areas of influence of geometric quantities on the stress concentration factor are obtained:

Area of negligible influence:

- weld width

Areas of small influence:

- base metal thickness;
- reinforcement height;
- weld toe angle with the toe radius bigger than 1 mm;
- toe radius with the radius bigger than 1 mm;

Areas of significant influence:

- weld toe angle with the toe radius of up to 1 mm;
- toe radius with the radius from 0.5 mm to 1 mm;
- Area of great influence:
- toe radius with the radius of up to 0.5 mm.

The graphs in Figure A1 show the areas of significant and great influence of geometric quantities. It can be concluded that the toe radius has a great influence on the stress concentration factor, so that the optimization of the welding procedure will be directed to this geometric quantity. The graphs in Figure A1 also show the influence of the toe radius change, that is, the influence of the change of the weld toe angle on the stress concentration factor. They were obtained according to expressions (A3b) and (A3c).

For a better understanding of the graphs in Figure A1, points "A" and "B" are selected to be specially analyzed herein. Point "A" is shown on the graph that presents the effect of the change of the toe radius, while the value of $U\varphi$ is −53.71. The value is negative, which means that by lowering the toe radius, the stress concentration factor increases. The value of −53.71 is the coefficient of the direction of the tangent Equation (A2) in point ($\varphi = 0.1$mm, $t = 10$ mm, $W = 20$ mm, $h = 4$ mm and $\theta = 50°$).

The value of $U_\theta$ in point "B" is 16.87. It is found in the Figure A1 that shows the influence of the change at the weld toe angle. The value is positive, which means that by increasing the weld toe angle, the stress concentration factor is increased. The value 16.87 is the coefficient of the direction of the tangent in the Equation (A2) in the point ($\varphi = 0.1$ mm, $t = 10$ mm, $W = 20$ mm, $h = 4$ mm, and $\theta = 5°$).

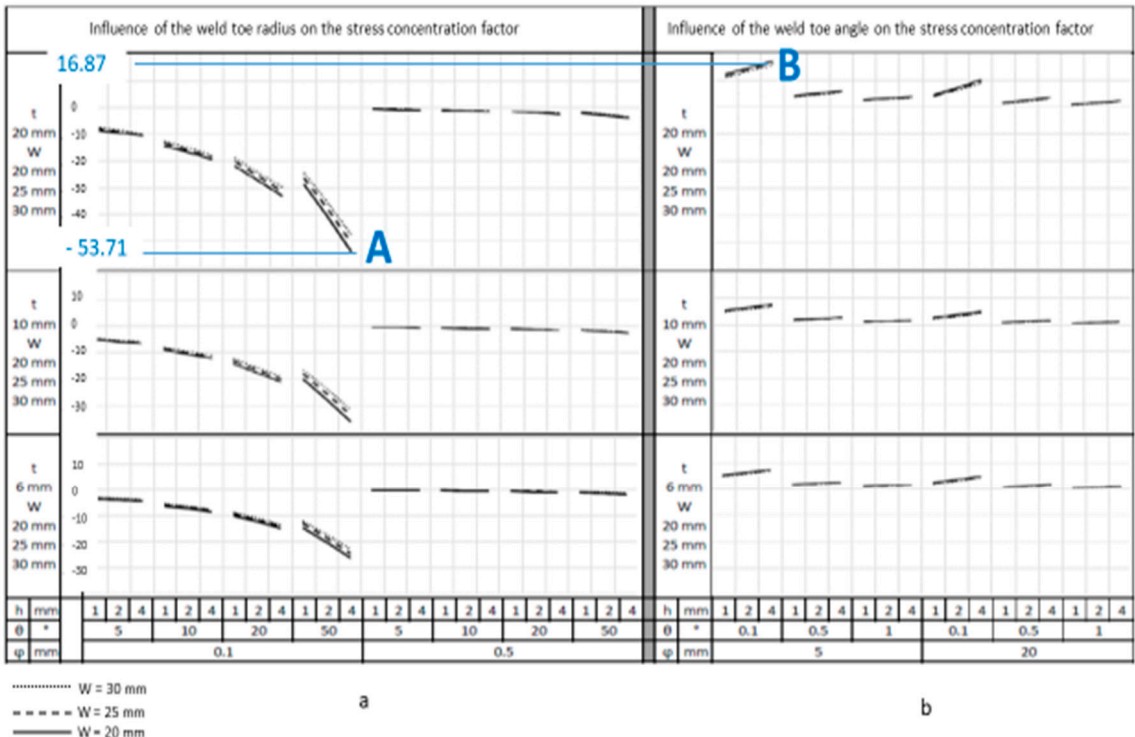

**Figure A1.** Areas of significant and great influence of geometric quantities on the stress concentration factor; (**a**) Influence of the weld toe radius on the stress concentration factor, (**b**) Influence of the weld toe angle on the stress concentration factor

Model B—Derivation of the Expression by Toe Radius and Weld Toe Angle Only

In the model B, the analysis of influence of geometric quantities was carried out in two steps.

In the first step, values of geometric quantities that are expected during the experiment are included in the expression (A2). For each geometric quantity the analysis was carried out with seven values:

The thickness of the base material (6, 10, 14, 18, 22, 26, and 30 mm);

The reinforcement height (1, 2, 3, 4, 5, 6, and 7 mm);

The weld width (2, 5, 8, 11, 14, 17, and 20 mm) and

The weld toe angle (10°, 20°, 30°, 40°, 50°, 60°, and 70°).

The toe radius was analyzed for the values from 0 to 12 mm.

Diagrams in Figure A2 show the influence of the thickness of the base material, the reinforcement height, the weld width, and the weld toe angle on the stress concentration factor with regard to the change of the toe radius. The analysis was made for seven values of each geometric quantity.

From the aforementioned diagrams it can be concluded that the reinforcement height and the weld width have a very small influence on the stress concentration factor. The thickness of the base material is constant so that there is no use analyzing the influence of the thickness of the base material. This is why we will continue by making the comparison of the influence of the weld toe angle and the toe radius on the stress concentration factor. In the further analysis, three geometric quantities will be fixed, namely, the thickness of the base material to 10 mm, the reinforcement height to 2 mm and the weld width to 18 mm.

In the second step, the sensitivity analysis was carried out with regard to the two geometric quantities that in the foregoing analysis were established to have the biggest influence on the stress concentration geometric factor. The sensitivity analysis was carried out in order to determine the level of impact of each single geometric quantity on the stress concentration factor and to direct the further optimization of welding parameters. It was carried out in a way that the Equation (A2) was partially

derived with regard to the two geometric quantities for which it was earlier established to be of the biggest influence on the stress concentration factor, i.e., Equations (A3b) and (A3c).

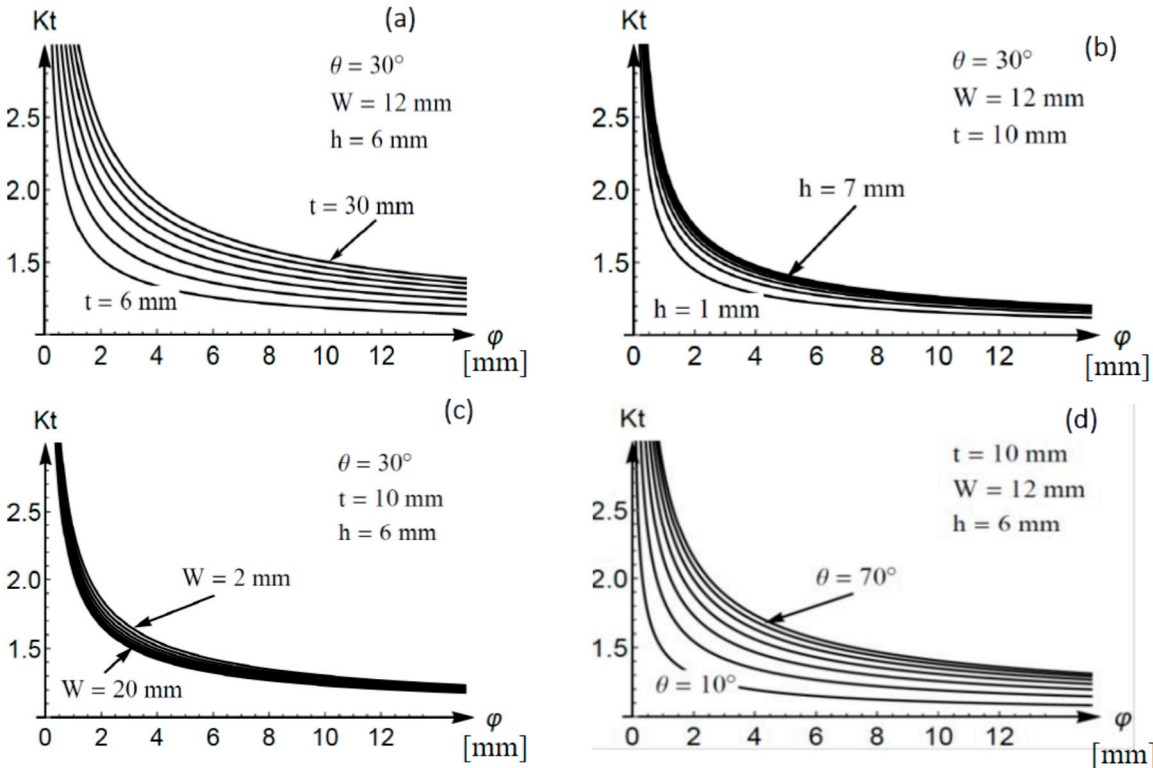

**Figure A2.** The analysis of the influence of geometric quantities (**a**) thickness of base material, (**b**) reinforcement height, (**c**) weld width and (**d**) weld toe angle on the geometric factor of stress concentration as function of toe radius after the expression suggested by Ushirokawa and Nakayama [28].

Graphs in Figure A3 show the influence of the toe radius (a) and the weld toe angle (b) on the stress concentration factor. Two points, "A" and "B", were selected in diagrams in order to have a better understanding. The value of point "A" is $U\varphi = -21.48$. This point is in diagram a, which presents the influence of the toe radius on the stress concentration factor. The value is negative, which means that by lowering the toe radius, the stress concentration factor increases. Value $-21.48$ is the coefficient of the tangent expression (A2) in point ($\varphi = 0.1$ mm, $\theta = 60°$, $t = 10$ mm, $h = 2$ mm, and $W = 18$ mm).

The value of point "B" is $U_\theta = 4.14$. This point is found in diagram b, which presents the influence of the weld toe angle on the stress concentration factor. The value is positive, which means that by lowering the weld toe angle, the stress concentration factor is increased. Value 4.14 is the coefficient of the tangent expression (A2) in point ($\varphi = 0.1$ mm, $\theta = 60°$, $t = 10$ mm, $h = 2$ mm, and $W = 18$ mm).

We can conclude that it is the toe radius that has the biggest influence on the stress concentration factor. With the analysis of the results obtained by the carried-out experiments, it will be established which welding parameters have the biggest influence on the toe radius in which the smallest stress concentration will be generated.

Appendix Conclusion

It can be concluded that the toe radius has the biggest influence on the stress concentration factor, especially for very small values of the radius. When changing the toe radius for values smaller than 0.5 mm, the stress concentration factor changes significantly. During the welding of butt welded sheets it is definitely necessary to avoid the radius smaller than 0.5 mm as in a such a small radius big stress concentrations take place, which can initiate the occurrence of surface cracks.

The influence of the weld toe angle is big, however, it is not as significant as the influence of the toe radius.

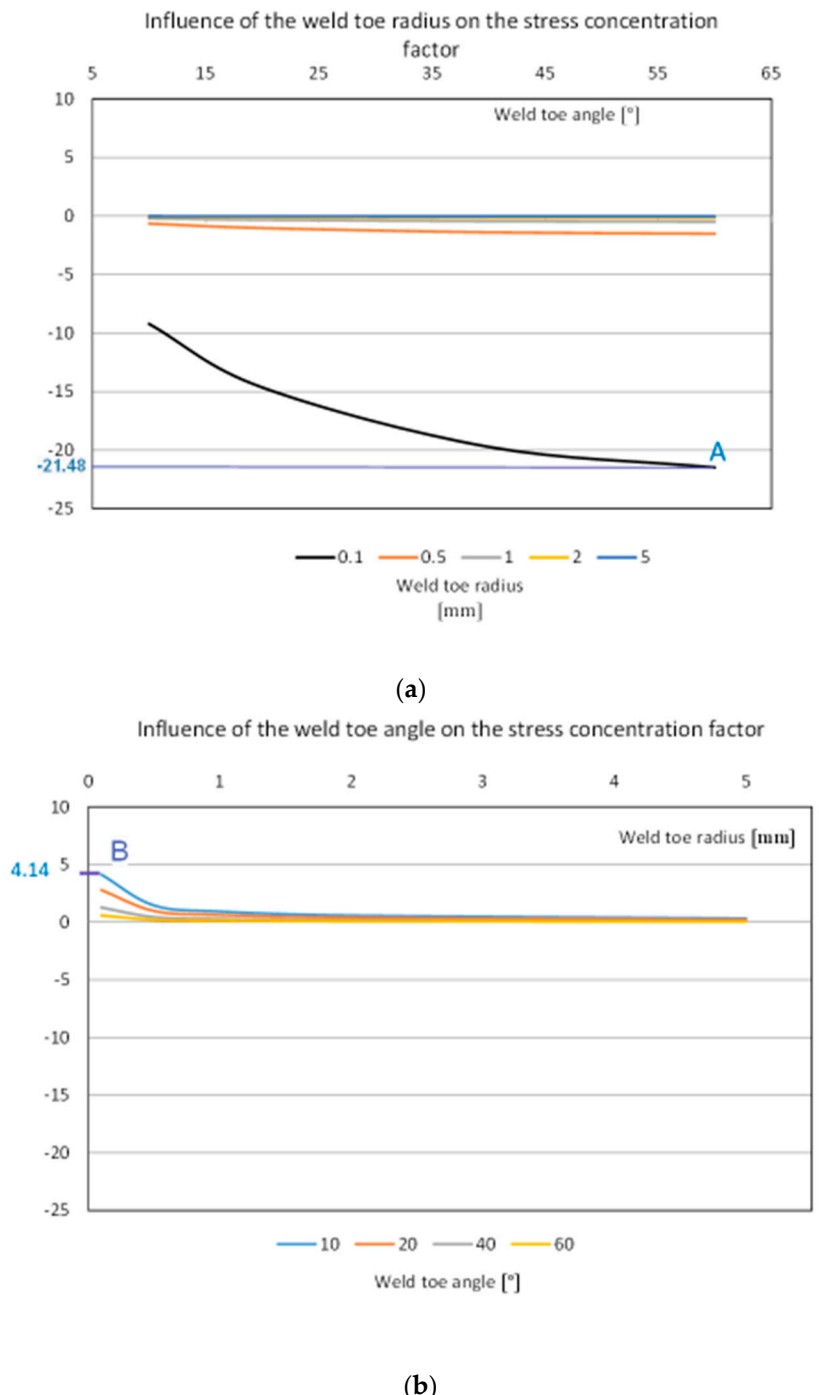

(**a**)

(**b**)

**Figure A3.** (**a**) Influence of the weld toe radius on the stress concentration factor, (**b**) Influence of the weld toe angle on stress concentration factor.

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
