# Peer review of "Multiparametric Investigation of Welding Techniques on Toe Radius of High Strength Steel at Low-Temperature Levels Using 3D-Scanning Techniques"

_metals, doi:10.3390/met9121355_

Round 1

Reviewer 1 Report

The present work is devoted to the study of the influence of process parameters on the toe radius of welded joints. The work is intersting and scintifically written but some modifications have to be performed before publication. Considering this i suggest major revision before publication.

-State clearly in the introduction the obectives of the work and the novelty in comparison with the literature

-The materials and methods part is too long. I suggest to move the explanation of the effect of the different process parameters in the introduction limiting the materials and methods to the explanations of the techniques employed and the different parameters studied.

-In the results part is too short. In general i suggest to revise the structure of the manuscript moving parts from the materials and methods to the introduction and to the results section.

-Add to the results section at least one example of the measure with the 3D scanner and not only the statistical analysis.

-Fig.10 and 11 are not readable. You have to enlarge the captions and the legend and the written parts of the figures

-The discussion must be more deep, with a comparison of the results with the ones from litarture and trying to explain why a certain behaviour was found

Reviewer 2 Report

The papers is well presented and describes an interesting and relevant topic. Improvements needed are mainly related to English language, as noted:

line 17: the sentence is not clear: "Though the shielding gas has a great influence on the toe radius, it is not significant."

line 292: legend of table is duplicated

line 309: table 5: some p-values are highlighted in red, it is not clear why

line 343, 354 and elsewhere: "biggest" should not be used and replaced by "higher" or similar formulation

line 359: The discussion section is very short. It should not use language as "we do not recommend" (first person)

line 360: "done" should be replaced by "performed"

line 365: "bigger" should be replaced by "higher"

line 377: "... big, although not a significant..." (formulation not clear)

Reviewer 3 Report

The paper has presented the effect of welding parameters on the toe radius of butt joint. It is more likely to be a lab report contains only results obtained by a 3D scanner without any scientific content/improvement. It is not suitable for publication in a good quality journal like Metals. The discussion should contain the reason behind the different toe radius depending on the welding parameters.

Round 2

Reviewer 1 Report

Considering that the authors have ansered to all the main issues, the article can eb accepted in its present form

Reviewer 3 Report

The text has been improved.